# Navigating Life with HIV as an Older Adult in South African Communities: A Phenomenological Study

**DOI:** 10.3390/ijerph17165797

**Published:** 2020-08-11

**Authors:** Naomi Hlongwane, Sphiwe Madiba

**Affiliations:** Department of Public Health, Sefako Makgatho Health Sciences University, P.O. Box 215 Medunsa 0403, Pretoria 0001, South Africa; naomihlongwaneot@gmail.com

**Keywords:** South Africa, older HIV-positive adults, disclosure, stigma, acceptance, phenomenology, navigating health care

## Abstract

The study explored how older adults experience an HIV diagnosis, deal with issues of stigma and disclosure, and navigate the healthcare system. Descriptive phenomenology was used to collect data from 20 older adults receiving antiretroviral treatment in health facilities in Gauteng Province, South Africa. Data analysis was inductive and followed the thematic approach. After diagnosis with HIV, the older adults experienced shock and disbelief, internalized their new reality of being HIV-positive, and found it difficult to disclose their HIV status. Stigma was embedded in their patterns of disclosure, and they chose not to disclose in order to manage stigma, protect their status, and maintain their privacy. Moreover, nondisclosure allowed them to achieve some normality in their lives without the fear of rejection. The older adults adopted various survival skills that aided them to accept their HIV status. Self-acceptance was important for the older adults who did not want to feel cheated out of life by the HIV diagnosis. The positive supportive attitudes of the healthcare professionals provided the much-needed support network for the older adults immediately after they had received their HIV test results. This was instrumental in their acceptance of their HIV status, their adherence to clinic visits, and their ability to live a healthy, positive life.

## 1. Introduction

HIV remains a global epidemic with an increased number of individuals aged 50 years and older living with HIV. Adults aged 50 years and older are now being acknowledged as a substantial proportion of the HIV-infected population globally. This is due to the increasing number of older HIV-positive adults [1]. As suggested by the United States Agency for International Development (UNAIDS: Geneva, Switzerland), the number of people living with HIV (PLHIV) 50 years and older tripled from 2000 to 2016. In terms of this trend, modelling predicts that, by 2040, this proportion of the population will increase to 27% in sub-Saharan Africa (SSA) and the number of HIV-infected adults aged 50 years and older will increase to 9.1 million [2]. Globally, this number is expected to increase to an estimated 7.5 million by 2020, with Eastern and Southern Africa being the epicenter of this epidemic with an estimated 2.9 million PLHIV 50 years and older [3].

Research has suggested that older HIV-positive adults may experience living with HIV differently than those who are younger. Older HIV-positive adults face an increased risk of experiencing a combination of HIV-related stigma and ageism [4]. In addition to the fear of rejection and social isolation, the fear of stigma is related to the social consequences of HIV-related symptoms, which often result in a limitation of social contacts, and this adversely affects the sources of support [5,6]. The sources of support for older adults are already inadequate as they are often single, live alone, or have fewer friends [3].

Research shows that after having received news of an HIV-positive diagnosis, acceptance amongst this group follows two pathways. Those that have accepted reported that they feel good about themselves, that they are at peace with themselves, and that they have found meaning in their lives, which they have shared with their families [7]. Whereas others, due to their poor support structure, had poor health-seeking behavior [8]. After diagnosis, they find it a challenge not only to comprehend the HIV diagnosis at their age, but also to navigate the HIV care services in primary health facilities [9].

Patterns of onward disclosure lie at the centre of how one is supported and adheres to antiretroviral therapy (ART). Research shows that older HIV-positive adults delay or rarely disclose their HIV-positive status to the significant others in their lives. Older adults in a study conducted in Uganda delayed disclosure for fear that their spouse/partner would leave them if their positive status was known [10]. The HIV- and age-related stigma, whether real or perceived, hinder disclosure; older adults avoid or delay disclosure to avoid rejection and gossip [11,12]. Among older HIV-positive adults, disclosure was significantly associated with the time since receiving the HIV diagnosis, the use of HIV services, having a confidant, age, having a steady sexual partner, and marital status. These were predictive factors for disclosure among older persons [13]. Older HIV-positive adults delay disclosure due to fear of stigma, and in order to manage potential stigma.

Little is known about the health or well-being of older HIV-positive adults who may be infected by HIV. Moreover, sero-conversion among this population group suggests that adults aged 50 years and older engage in activities and behaviors that put them at risk of HIV transmission [14]. A previous South African, found that older men in particular accept engaging in multiple sexual partnerships [15]. With the growing number of older adults living with HIV, it is imperative to understand the contextual features that shape how they experience an HIV diagnosis, their patterns of onward disclosure, and how they navigate HIV services in primary health care facilities.

## 2. Materials and Methods 

### 2.1. Study Design and Setting

This was a facility-based study conducted from April to July 2018 in primary health care facilities in Tshwane District region 1. The District is located in the City of Tshwane, a metropolitan municipality with a population of about 3.3 million, in Gauteng Province, South Africa. The elderly constitute 24% (1.18 million) of the population in Gauteng province, and approximately 811,570 elderly people reside in the region [16]. There are 24 health facilities that offer ART services and a range of other health care services within the region. The facilities offer integrated and non-integrated services to clients from urban, peri-urban, and informal settlements that make up 21% of the population in the District. Three facilities were selected for the study. They are located in peri-urban areas that have been populated by informal settlements and accommodate a multi-cultural population living in sub-standard conditions.

The participants in the study consisted of HIV-positive adults aged 50 years or older who were accessing ART services in the selected facilities. To be eligible for participation in the study the participants had to be diagnosed with HIV within five years at the time of the study. Descriptive phenomenology was used to explore and gain an in-depth understanding of how newly diagnosed older HIV-positive adults experience an HIV diagnosis as well as how they live with HIV. Phenomenology was used to understand and describe their lived experiences without trying to predict or explain their behavior [17,18]. The participants were selected using purposeful sampling, which is consistent with a phenomenological enquiry. Purposeful sampling involves the selection of information-rich cases for in-depth study of the phenomenon under investigation [19,20]. In this study, older HIV-positive adults who had a recent HIV diagnosis were selected. The sample consisted of 20 older HIV-positive adults, because the goal of phenomenological research is not to create results that can be generalized, but to gain insight into meanings attached to the experience of the phenomenon. The sample size was thus influenced by its ability to provide rich, in-depth data portraying individual experiences of a recent HIV diagnosis. While the goal of phenomenology is to select a homogeneous sample, variability and diversity in terms of socio-demographics such as age, marital status, educational status, and living arrangements were pursued to ensure variety in the information, and thus to reduce information bias [21,22].

### 2.2. Data Collection

Face-to-face interviews were conducted, by the lead author and a trained research assistant, over a four-month period. Semi-structured interviews were performed, using a self-developed interview schedule with open-ended questions and possible probes. All interviews were conducted in the local languages, this is consistent with phenomenological inquiry [23]. As a result, the interview schedule was translated from English into the local languages (Setswana and IsiZulu) to allow the participants to speak freely. The participants were asked four broad open-ended questions about (1) the experiences of testing HIV-positive as an older adult, (2) challenges about disclosing their HIV status, (3) the kind of support they received after being diagnosed, and (4) their experiences of accessing health care facilities [18]. In phenomenology the researcher describes the participants’ experience in the way he or she experiences it, and not from some theoretical standpoint [23]. In addition, other follow up questions and probes were asked in response to the responses given. This allowed for a dynamic exchange between the researcher and the participants, to obtain descriptions of context and elicit meaning. The participants were interviewed only once each. The interviews were on an average 45-min long and were recorded with the consent of the participants. The participants were interviewed in consulting rooms in the health facilities for privacy. The participants were informed about the voluntary nature of their participation and their right to withdraw from the study at any stage.

### 2.3. Data Analysis 

All interviews were first transcribed verbatim in the local language, and then translated into English by the lead author and the research assistant who was involved in the data collection. The authors checked the transcripts for accuracy against the audio recording. The analysis of the data followed a inductive thematic process that is embedded in the nature of descriptive phenomenology [24]. Each transcript was read several times by the authors, line by line, so that they could immerse themselves in the data and identify the initial codes emerging from the data, thus developing a codebook. The authors held many sessions to revise the codebook and to reach consensus on the emerging themes and subthemes. The final codebook was used to apply codes to all the transcripts using NVivo [25], a qualitative data analysis package, which was utilized for the analysis process. The authors then explored and extracted statements for their meaning in order to realize emerging themes that described these lived experiences. Quotes from the interviews were used to elucidate the lived experience. 

To ensure rigor and trustworthiness, we used several strategies to satisfy credibility, transferability, dependability, and confirmability as described by Guba and Lincoln [26]. We conducted the interviews in local languages, conducted several sessions of peer debriefing, and used a good digital recorder to enhance verbatim transcription of the responses to reflect the participants’ experiences of the phenomenon. The lead author maintained a reflective attitude throughout the research process to reducing researcher bias [27], and kept a detailed audit trail with thick descriptions of the research processes and procedures. In addition, both authors analyzed the data to reduce the effect of investigator bias and ensure that the interpretations were free from bias and the conclusions credible. In phenomenology, rigor is also ensured through the thoroughness and completeness of the data collection and analysis [28], which further satisfy the prolonged engagement with the field. 

### 2.4. Ethical Considerations 

Ethical clearance for this study was obtained from the Research and Ethics Committee of Sefako Makgatho Health Sciences University (SMUREC/H/111/2017: PG) and the Management of Tshwane Central Sub-district. Participants were told that their participation was voluntary. All provided written informed consent before the interviews. Pseudonyms were used in presenting the quotations to ensure privacy and anonymity.

## 3. Findings

### 3.1. Participant Context

Table 1 presents the socio-demographics of the participants. They were adults aged 50 year or older who had recently been diagnosed with HIV. All participants were on ART at the time of data collection. Most (11 out of 20) had been on ART for 3–5 years, and only 9 had been on ART for less than 2 years. Their age ranged from 50–65 years. Most (9 out of 20) were above 60 years of age, while only three were aged 50–55 years. Females accounted for 15 out of the 20 participants. Most of them (12 out of 20) had chronic illnesses, and hypertension accounted for 8 of the 12 with chronic illnesses. All had selectively disclosed their HIV diagnosis to someone.

### 3.2. Emergent Themes

Figure 1 is a thematic representation of the themes that emerged during the analysis of the in-depth interviews. These were (1) the reaction to learning about the HIV diagnosis, (2) navigating life with HIV, (3) the dimensions of disclosure, and (4) accessing health care.

#### 3.2.1. Reaction to Learning about the HIV Diagnosis

The participants’ responses to their HIV diagnoses were complex and influenced not only by their age, but also by their low risk perceptions for HIV infection. The data revealed that though the participants were tested for HIV due to the presence of symptoms, they had not associated their symptoms with the possibility of being HIV-positive. Three sub-themes emerged: Fear and pain, expectancy and acceptance, and social isolation.

##### Fear and Pain

The participants expressed disbelief as to how they could be HIV-positive, especially because only four of them were living with their partners and had an active sexual life.


*I felt scared because it was something that I was not ready for… Yho…. (quietness). I just told myself that from now on I would have to start taking treatment so at first; I didn’t accept the news very well because it came as a shock to me but as time goes I then accepted.*
(Johanna, a 58-year-old female)


*Yho (sigh). I felt hurt and I wondered because my husband died and it’s now about 15 years since he passed on. They did not tell me what killed him, I thoughts that it’s just an illness. There is nothing I could do…, I told myself that I am not the only one and I will take care of myself.*
(Nancy a 55-year-old female)

##### Expectancy and Acceptance

Having a partner who is HIV-positive brought out a different initial reaction, as shared by Sam, a 63-year-old male whose wife is HIV-positive. His narrative reflected a sense of expectancy and acceptance.


*As long as I know that my wife has been positive for long, and the time she was positive I never had that thing that I must go (leave her) because she is positive. I told myself that I accompanied her to test and she started medication. I was supporting her and when it appeared to me that I am positive, I did not have a problem.*


##### Social Isolation

Upon discovering their HIV-positive status, some participants resorted to isolating themselves from social settings. While isolation after finding out their HIV status might result from not knowing how to deal with being positive, for participants in the study, social isolation was used a way of avoiding rejection because of their HIV-related symptoms. 


*Ahh, no on my side they gossip and I don’t care about people. I don’t go visiting in people’s homes. I sit alone in my house.*
(Shiela a 63-year-old female)


*Yho... why do I need to stress myself by going around visiting friends and neighbors because you know the people in the township talk too much so I stay home with my family.*
(Martha, a 53-year-old female)

#### 3.2.2. Navigating Life with HIV 

The study explored how the participants navigate life as older persons living with HIV. Although becoming HIV-positive changed the lives of the participants, their narratives showed that they became resilient and developed coping mechanisms in order to positively navigate the illness. The theme of navigating HIV contained four sub-themes: keys to survival, abstaining from sexual relations, accessing social support, and managing stigma. 

##### Keys to Survival

The participants discussed various processes that helped them to survive the HIV diagnosis and to come to terms with it. The data revealed that they employed various approaches to coping: not feeling cheated by the disease, the need to accept in order to cope, trust in ART enabling adherence, and accessing support.


*The need to accept in order to cope*


Accepting the HIV-positive status is critical part of living with HIV. The participants admitted that self-acceptance was an important process if they were to begin to cope with their reality of being HIV-positive at their age. 


*You know I just asked myself how come …, what happened to me to have HIV but still I didn’t find an answer to that question so I just told myself I had to accept there is no other way you know.*
(Olga, a 52-year-old female)


*I have accepted and it [HIV] has become my friend and it’s just a virus. I don’t think it’s going to kill me. I will die when I want to die and provided that God has taken out my card, HIV is my brother.*
(Malifu, a 59-year-old female)

Acceptance took place against a backdrop of conflicting emotions and endless questioning, and became a necessity in moving on. 


*Living with HIV as an elderly person, it is a bit difficult. You know you feel like you are being looked down upon. And that if I am coming to collect medication (ARV’s) at my age, people take it that I am sleeping around. And when you think that you are taking care of yourself, you have one partner, it’s difficult to understand. And you just accept because there is nothing else you can do, but when you ask yourself that because I once tested and I was negative now why am I positive, how did this happen? That really bothered me. I asked myself these questions and could not get the answer as to how this happened.*
(Nora a 64-year-old woman)


*Not feeling cheated by the disease*


Knowledge of their HIV-positive status was reported by the participants to give rise to feelings of guilt, blame, and a sense that one’s life had been cut short. However, the data revealed that the participants did not feel cheated by the disease:
As I have explained, I have accepted my life. I don’t perceives myself…, meaning me being HIV-positive, I don’t take it as being a problem. If I can take care of myself I will be OK just like the person who is negative.(Marriam, a 65-year-old female who has been HIV-positive for one year)
As I said before that I have accepted. Now I’m used to the idea of going to the clinic to get my treatment, and it is something that people are used to by now. It’s like a person who is diabetic.(Johanna, a 58-year-old female)


*Trust in ART enabling adherence*


The participants expressed a strong trust in the ART medication. They believed that it was possible to live a healthy life because of the advances in the ART treatment.


*They [the nurses] said I should not worry. There was no pill back then, and it [HIV] was killing people. But, now it’s better because I can live in my age and not show any signs that I am sick. The pill is treating me well. And even when you tell someone that you are on the pill they won’t believe. I don’t give myself stress that I am sick, I accept everything that happens in my life.*
(Sheila, a 63-year-old female)


*They [the nurses] said if you drink this medication as prescribed, you will live a long life. So from that day I told myself that since I want to live and I don’t want this illness to bring me down, I will surrender myself to the treatment and take it at the time they say I should.*
(Nora a 64-year-old woman)

##### Accessing Social Support 

Social support for older adults diagnosed with HIV has been linked with the reduction of HIV-related stigma and as contributing to living positively with HIV. The participants said that they sought and received support from their families, friends, and healthcare workers at the clinics they attended. They described the support they received from their family members as a source of comfort that was critical in helping them successfully cope with the HIV diagnosis. 


*She [the sister] never reacted funny. She was actually the one encouraging me, even though she did not understand how did this happen because she knows what kind of a person I am. But she supported me. She was the one who brought me with her car, supporting me, when I came here for the first time, as I was sick at that time.*
(Nora, a 64-year-old woman)


*They [the family] give me courage and always ask me if I have taken my medication yet. Like today when I came to the clinic, I live with my mother, so she asked when am I going to the clinic.*
(Samuel, a 59-year-old male)


*The nurses give me courage that if I can continue taking my pills and not engage in unsafe sex and use protection so that I should not infect others.*
(Enos, a 59-year-old male)

##### Abstaining from Sexual Relations 

Older persons are likely to experience difficulties regarding intimacy and sexuality after an HIV diagnosis, particularly older menopausal women who experience a drop in sexual interest. Most (16 out of 20) participants in the study reported absence of a sexual life at the time of data collection. The three main issues that the participants discussed in relation to their sexual relations post HIV diagnosis. 

Widows who associated their husband’s death to AIDS refrained from any sexual activity. 


*When I was fist told that I am positive, I told myself this thing called a man I don’t want to see it in my life and will not find myself sleeping with one until today.*
(Nancy a 55-year-old female)

On the other hand, those in relationships found difficulty negotiating sudden condom use and refrained from any sexual activity.


*My husband and I have not had sex all together up to now from the time I started treatment (started ARV;’s a year ago) until now.*
(Marriam a 65-year-old female)

Despite the HIV diagnosis, the male participants still wanted to be sexually active, but refrained due to the fear of spreading the virus.


*I just dealt with it-even sex. I don’t do sex anymore and is not because I don’t crave it. I do! Especially after I took my medication I crave for sex.*
(Samuel, a 59-year-old male, HIV-positive for six months)

##### Managing Stigma

The stigma associated with HIV and seropositivity disclosure surfaced as a significant challenge for our participants, whether this stigma involved the fear of rejection, the actual rejection experienced after disclosing seropositivity or the difficulty of disclosing one’s HIV status in an intimate context. But they found ways to manage stigma.

Not fearing HIV-related stigma was a way of managing its impact. 


*When I came to test for the first time I didn’t care who was watching me and saying “What is that old lady doing here?” I came in to test …, I didn’t care about what people would say. Many of us [older people who collect ART] know each other. Even when I collect my pills, I don’t care who says what. I take them and leave.*
(Flora, a 60-year-old female)

When stigma was experienced, the participants were robust, as they continued with life despite it.


*People do gossip about me, more so the time when I was sick. Skinny, they were saying. Look at her she will die She has AIDS. But I didn’t care I took it as though I was injured, but they saw me as being sick when I walked in the streets because I was very skinny, and they would just talk bad about me.*
(Margret, a 54-year-old female, HIV-positive for one year)

#### 3.2.3. Dimensions of Disclosure 

The narratives with HIV-positive older adults revealed different dimensions of disclosure that were influenced by personal experiences of violation of their confidentiality, an attempt to maintain control over who knew about their HIV status, and the need to minimize unintentional disclosure. Three commonly identified styles used by the participants as they navigated disclosure to significant people in their lives included selective disclosure, protecting their HIV status, and not disclosing.

##### Selective Disclosure

In disclosing their HIIV status, some HIV-positive older adults combined intentional and selective disclosure. They disclosed intentionally to selected people, because of the relationship they had with the individuals. These included their spouses, siblings, friends, or children. Selective disclosure for some of the HIV-positive older adults emanated from knowing the individual’s HIV-positive status.


*For me the person I told was my partner, and before I told him it was difficult for me at first, but because I live with him and he married me and he is my partner he was the first person I told. I told him that I am positive, but he accepted and he supports me.*
(Nora, a 64-year-old female)


*It was not difficult; it was easy. As I mentioned that in my family there is already someone who is HIV positive. I didn’t hide my status from them, and they just accepted me.*
(Olga, a 52-year-old female)

##### Protecting One’s HIV Status 

The narratives with HIV-positive older adults revealed that considerations regarding disclosure involved the need to protect their HIV status, which also entailed protecting information about their medication. They expressed anxiety over possible unintentional disclosure whereby people might link the medication to their HIV status, and described how they made sure that their medication was hidden from some of the members of their households. 

Marriam, a 65-year-old female who has been LHIV for one year, who lives with her children and who has not disclosed to any of them said:
Not that I hide them [the medication], but I put them where I know they cannot reach.

Catherine, a 62-year-old female who has been LHIV for three years and only disclosed to one child out of her four children had this to say:
I hide them [the medication] from the children, but I do not forget to take them. They don’t make me sick; nothing at all. Even when I go to places, my daughter will say Mama, are you leaving? You didn’t forget …, and she will call me to the corner and whisper to me “your pills”.

##### Not Disclosing to Manage Stigma

Some participants had chosen nondisclosure, because of their fear of what might result from disclosing their status to anyone, including close family members. They talked about their fear of being rejected as an important reason for choosing not to disclose. 


*According to me I would not disclose to other people because when you have this illness they exclude you and ignore you, so it’s better to keep quiet.*
(Norman, a 55-year-old male)


*I told my daughter about the HIV and we are OK, but I have not told the boys, because you know these boys get drunk, and soon he will be swearing at you on the streets.*
(Maria, a 57-year-old female)

##### Violation of Confidentiality

For some of the HIV-positive participants, disclosure resulted in a violation of confidentiality by those to whom they had entrusted their secrets, who subsequently disclosed their seropositive status without permission. The narratives revealed that the violations of confidentiality came from sources such as friends and family members. 


*I disclosed to my neighbor, and when we fought, my neighbor told my husband that I was HIV-positive. But my husband did not confront me and asked me why did I not tell him that I was positive.*
(Marriam, a 65-year-old female who has been LHIV for one year)


*I told my sister who is a nurse that I am positive. She was like, No problem, my sister. Just take your medication and you will be ok. But when I went to visit my mother, she asked me why didn’t I tell her that I was sick, which means that my sister told my mother. Since then I have not told anyone.*
(Joyce, a 53-year-old female who has been LHIV for a year)

#### 3.2.4. Accessing Health Care

The participants’ narratives showed that they adhered to their regimen of clinic visits, despite challenges pertaining to the accessibility and availability of services and resources. The distance travelled to the clinic was a barrier to accessing health care services. 


*You can sit here [in the clinic] the whole day while they say they are looking for your file and end up not finding it. They then send you back home on foot because you do not have money for a taxi. When you come back the following day, they still cannot find your file. And you travel by foot and you find you don’t have money for food and you go back hungry.*
(Sheila, a 63-year-old female who has been HIV-positive for eight months)


*It’s a burden. Even now I am just thinking if only they could give me medication for the next three months so I can rest a bit. It’s a long distance to come here. You wake up very early.*
(Nancy, a 55-year-old female)

The unavailability of resources made participants despondent.


*Sometimes you come to the clinic and they don’t have needles…. They then tell you to come the following day, and when you get to the clinic there are still no needles. It can take a whole week and there still will be no needles. Then it becomes a problem at your work and people say this person is always at the clinic.*
(Sam, a 63-year-old male who has been HIV-positive for three years)

Healthcare providers and other clinic attendees provided a support network for the participants and aided in giving them a positive experiences of the health care services.


*They treat us well… I am satisfied and I think if I was sitting at home, the information that they have here [in the clinic] I would not get from anyone.*
(Marriam, a 65-year-old female)


*I meet other older people who come to collect their pills. We don’t have any problems. We end up having conversations here in the queue.*
(Gugu, a 60-year-old female)

## 4. Discussion

The purpose of this qualitative study was to explore how newly diagnosed older persons experience an HIV diagnosis, to assess their patterns of onward disclosure, and to explore how they navigate HIV services in health facilities. The study found that HIV testing is delayed in older persons due to their low self-perceptions of the risk of HIV transmission and acquisition. Most had never used a condom during sexual intercourse before the HIV tests, had had multiple sexual partners, and were not greatly concerned about the risk of HIV infection. They had tested for HIV due to their experiencing of unfamiliar symptoms. The observations are consistent with those in other studies [3,29]. The lack of awareness by healthcare providers of the HIV risk among adults aged 50 years and older affects their HIV testing patterns [30]. This results in older adults being diagnosed with HIV at a more advanced stage of the disease [1]. There is a need to focus HIV-related and prevention messaging and programs for older adults.

They had been shocked upon hearing of their HIV-positive test results, with most reflecting and asking themselves how they could be HIV-positive at their age. The data further revealed that they lacked interest in engaging in sexual intercourse and had difficulty in resuming a sexual relationship, after testing HIV-positive. In line with previous findings, some were frightened of infecting someone else [31]. The study also found that some of the participants had resorted to socially isolating themselves, as a way to prevent being stigmatized. In the current study, social isolation co-occurred with fears of perceived stigma, where the participants socially isolated themselves, in order to mitigate the effects of stigma. In line with findings from previous studies, social isolation was particularly common when AIDS-defining symptoms were visible, and the fear of stigma was linked to the fear of rejection [32]. Previous and recent findings show that HIV-infected older adults are significantly more isolated than their younger counterparts are, and that their support networks are inadequate [4,33]. It is important that health facilities develop interventions to target HIV-related stigma to reduce loneliness in older adults living with HIV. Support groups could play a crucial role in reducing loneliness and creating a safe environment where PLHIV can disclose safely and openly talk about their fears.

In the context of the HIV diagnosis, the participants employed various survival strategies that aided them in navigating life as older persons living with HIV. Most were aging positively, despite their HIV status. They did not feel cheated by the HIV diagnosis and adopted it like any other chronic illness. This observation is consistent with the observations made in a study of HIV-positive older persons in an affluent society. Emlet et al. [34] found that older persons living with HIV exhibit resilience and develop coping mechanisms to adapt to their condition over time. Having support and knowing someone who is HIV-positive assisted the older persons to cope with the HIV diagnosis [35]. Family, sexual partners, friends, family members, health care professionals, and other social groups were all regarded as sources of strength and support for older persons living with HIV. Similar findings have been reported elsewhere [7]. Furthermore, the narratives of the older persons indicated that they had a strong belief in the efficacy of the ART medication to help them survive the disease [34].

Self-acceptance is important in the lives of older persons living with HIV. It allows them to move forward with their lives after an HIV diagnosis. The current study observed a desire among the participants to accept their status as a means of coping. This observation is consistent with those reported in previous findings, where self-acceptance allowed the older adults to move forward with their lives after an HIV diagnosis [34]. The older persons in the current study needed to accept their status in order to be able to cope. Accepting is an accommodative way of coping [36]. However, the current study found that, even though the participants said they had accepted their status, there was an underlying tone of passive acceptance, as later they would express difficulties in disclosing and finding closure with their status. Research suggest that a passive attitude is associated with negative effects and faster disease progression in PLHIV [37].

The current study revealed that some participants had mixed feelings in deciding whether to disclose or not. Consistent with previous studies, they used nondisclosure as a stigma management tool [11,38]. They lived in multigenerational households, and when they learned about their HIV-positive status, they found it difficult to disclose it to some of the members of the household. They desired to protect their status and maintain their privacy, and went to great lengths to do so. This is evident in the way that they managed their ART medication; their narratives indicated that they hid their medication, in order to maintain this privacy. According to Emlet [4], not disclosing allows older adults to achieve some normality after testing positive. They can live without fear of rejection and stigma. Elmet’s [39] explanation applies to the participants in the current study who socially isolated themselves when they presented with AIDS-defining symptoms. They avoided rejection and gossip, which would shatter their sense of self-respect. 

The participants in the current study [40] were selective in their approach to disclosure in order to protect their status and guard against the spread of gossip and stigma. The data further revealed how stigma was embedded in these patterns of disclosure. As indicated, some participants chose not to disclose their status, in order to manage stigma and avoid being ostracized. Emlet [39] refers to the concept or phenomenon of protective silence as a mechanism for protection against potential HIV stigma in older adults. Research suggest an association between social isolation and non-disclosure [12].

Contrary to the frequently reported stigma amongst the older persons, an absence of the fear of stigma was reported by some of the participants in this study, which was in contrast to the findings of other studies investigating HIV-related stigma. The observation of low stigma scores and internalized stigma in older adults living with HIV was reported in previous studies [34]. Although this finding suggest that stigma declines with age, other researchers argue that older adults who do not experience or fear stigma use non-disclosure as a stigma management strategy [11]. Older adults who had not been discriminated against and had not experienced stigma had used selective disclosure in order to manage stigma [12]. 

Some participants had disclosed in confidence to significant people in their lives, but the confidentiality had been violated and the trust broken. This violation of trust had led the participants to a decision never again to disclose. This was particularly important to them, as the violation of confidentiality came from close family members or friends. Older adults in other settings also are expected to exhibit responsible behavior and guide younger people [41]. In the South African society, older persons have always felt a sense of responsibility to care for those in need and impart wisdom and advice regarding better life choices [42,43].

Navigating health services was not a new experience to the participants, who had previously used the services for chronic care, but it had its own dynamics after the HIV-positive test results. Most of the older persons (12 out of 20) had at least one chronic illness. Financial issues, the lack of transport, and the long distances they travel to the health facilities posed a challenge to them. However, the health care providers functioned as a support network for the older persons after they received their HIV test results, which helped them to adhere to their regimen of clinic visits.

## 5. Limitations of the Study

The study was limited by its qualitative design and its use of purposive sampling, which is inherent in all qualitative research. Another limitation was that the participants were recruited from health facilities that provide ART. The experiences of the older adults who had been diagnosed but were not receiving ART might be quite different. The authors acknowledge the intersection of gender and qualitative research and the limitations that might be the result of male-to-female (or vice versa) interview.

## 6. Conclusions

After the initial emotions of shock and disbelief, the participants internalized their new reality of being HIV-positive and found it difficult to disclose their HIV status. Stigma was embedded in their patterns of disclosure, and some of them chose not to disclose in order to manage stigma, protect their status, and maintain their privacy. Moreover, nondisclosure allowed them to achieve some normality in their lives without the fear of rejection.

The study found that the participants had adopted various survival skills that aided them to accept their HIV status. Self-acceptance was important for them, as they did not want to feel cheated out of life by the HIV diagnosis.

The positive supportive attitudes of the healthcare professionals provided the much-needed support network for the participants immediately after they had received their HIV test results. This was instrumental in their acceptance of their HIV status and their adherence to their regimen of clinic visits.

Health professionals should take cognizance of the dimensions and methods of disclosure identified in the study when they design counselling interventions for older adults who test and live with HIV. Furthermore, they should adopt the survival mechanisms used by older adults and incorporate these in developing support structures to aid in positive aging and the navigation of an HIV diagnosis.

## Figures and Tables

**Figure 1 ijerph-17-05797-f001:**
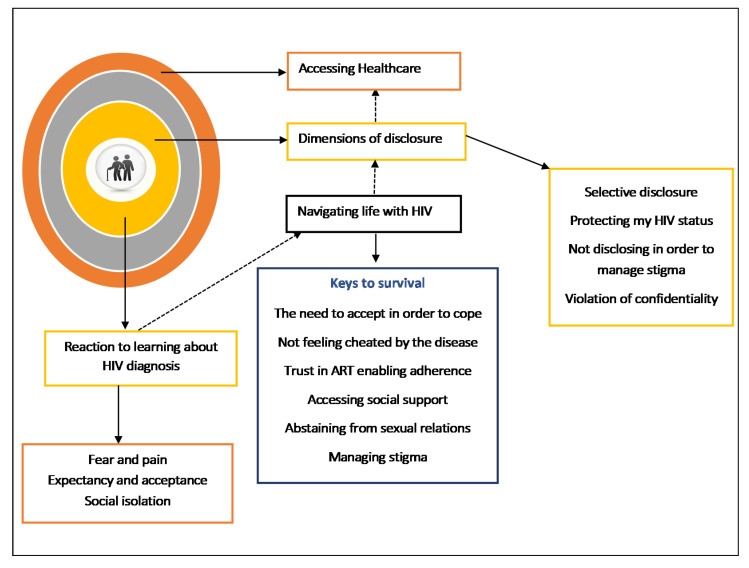
Summary of themes and sub-themes.

**Table 1 ijerph-17-05797-t001:** Sociodemographic and HIV-related characteristics of the participants.

Variables	Categories	Frequency	Percentage
Gender	Male	5	25.0
	Female	15	75.0
Age group	50–55	3	15.0
	56–60	8	40.0
	Above 60	9	45.0
Marital status	Single	3	15.0
	Married	11	55.0
	Widowed	6	30.0
Educational attainment	No formal schooling	3	15.0
	Primary education	4	20.0
	Secondary education and above	13	55.0
Employment	Employed	6	30.0
	Unemployed	14	70.0
Duration on ART	Less than 2 years	9	45.0
	3–5 years	11	55.0
Other chronic conditions	Yes	12	60.0
	No	8	40.0

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
