# Peer review of "Navigating Life with HIV as an Older Adult in South African Communities: A Phenomenological Study"

_ijerph, 2020, doi:10.3390/ijerph17165797_

Round 1

Reviewer 1 Report

This manuscript is well written. It contributes to the overall knowledge about the experiences of HIV in older adults in Africa. The introduction and methods sections are well written and highlight the significance of the problem and associated methodology. The description of strategies used to establish and maintain rigor are consistent with qualitative methods. The findings are well described and provide verbatim comments from participants to support the identified themes. The discussion of the results in comparison with current literature highlight the consistencies from this research as well as the gaps this study hoped to fill. Limitations are identified and implications for future research/practice with older adults with HIV in Africa are consistent with the extant literature and the study's findings. A few minor typos throughout should be corrected prior to publication.

Author Response

We thank the reviewer for their comments 

Reviewer 2 Report

This qualitative study described the experience of older adults living with HIV in South African communities, including the HIV related stigma, disclosure and healthcare experiences. This manuscript was well written and also has important public health implications. However, I have some methodological concerns as listed below although generally, it has a solid data analysis.

2.1 Study design and setting

In the second paragraph, the first sentence you mentioned “The participants in the study consisted of newly diagnosed HIV-positive adults …..” How did you define newly diagnosed? In Table 1, the variable “Duration on ART”, 11 participants have been on ART for 3-5 years. So you defined “newly diagnosed” for more than 5 years? That does not make any sense. Please explain. If you select newly diagnosed HIV patients, what is the justification? If the patients were still in the shock of HIV diagnosis, they don’t have time to get ready to disclose to anyone yet. I am also not sure how can they obtain social support without disclosing to other people.

Author Response

This qualitative study described the experience of older adults living with HIV in South African communities, including the HIV related stigma, disclosure and healthcare experiences. This manuscript was well written and also has important public health implications. However, I have some methodological concerns as listed below although generally, it has a solid data analysis.

Study design and setting

In the second paragraph, the first sentence you mentioned “The participants in the study consisted of newly diagnosed HIV-positive adults …..” How did you define newly diagnosed? In Table 1, the variable “Duration on ART”, 11 participants have been on ART for 3-5 years. So you defined “newly diagnosed” for more than 5 years? That does not make any sense. Please explain. If you select newly diagnosed HIV patients, what is the justification? If the patients were still in the shock of HIV diagnosis, they don’t have time to get ready to disclose to anyone yet. I am also not sure how can they obtain social support without disclosing to other people.

Response

Thanks for the comment, we changed newly diagnosed HIV-positive adults to “to be eligible for participation in the study the participants had to be diagnosed with HIV within five years at the time of the study. Disclosure to others is complex, some people disclose immediately, others delay disclosure, while some rarely disclose their HIV-positive status to the significant others in their lives. In a similar manner, HIV positive people obtain support from significant others. Those who delay disclosure to significant others often get support from the healthcare professionals who provide HIV care. The study also explored their experiences of engaging in care. The purpose of the study was to understand disclosure, support, and engaging in care as experienced by the participants throughout the care continuum.

Reviewer 3 Report

ijerph-853773

Navigating life with HIV as an older adult in South African communities: A phenomenological study

This descriptive Phenomenology study, analysed using inductive-thematic framework analysis, describes lived experiences of being with HIV among older adults. If the following comments are incorporated, the paper will contribute to HIV literature.

Abstract: None

Introduction

  • Line 36, make clear whether the number refers to the entire population of age 50+
  • Line 53-54, could the authors add more reasons of un-disclosure for this age group? Particularly related with family separation, job loss, etc….

Methods

  • Line 82, what was the purposive criteria? 15/20 were females which disproportionately seem to have the lived experiences by gender, for example.
  • Line 94-98, How to manage investigator bias as the lead author did the interview?
  • Line 98-102, what framework did the authors use to guide their in-depth interview? If not any limitation?
  • Line 25, could the authors provide some strategies to ensure credibility, transferability, dependability, and reliability of data

Results

  • Line 150, figure 2 resembles the socio-ecologic model such as access at macro (policy)-level, disclosure/stigma at meso (community)-level, etc. Was this co-incidence or the authors initially considered SEM as their framework? NB. I have raised comments related with framework earlier.
  • Line 150, figure 2, how the figure was constructed (how the themes and subthemes are interlinked) should be described somewhere.
  • Interpretation of quotes in the results section: I could not see much meta-phore interpretation of most quotes which could illustrate the deep phenomenology of the participants. I suggest the authors to interpret and add more. The journal does not have word limitation and they should use the opportunity.

Discussion

  • Line 385-391: What does the low self-perception and risky sexual behaviours such as not using condom, having multiple sexual partners and low concern to HIV infection tell out to the general population? What should the different stakeholders should do?
  • Line 404, any suggestion for the social isolation?
  • Any further limitations of investigator bias? Also, any limitation as a result of male-to-female (or vice versa) interview? Sometimes women may not describe their livid experience to men or vice versa?

Author Response

This descriptive Phenomenology study, analysed using inductive-thematic framework analysis, describes lived experiences of being with HIV among older adults. If the following comments are incorporated, the paper will contribute to HIV literature.

Introduction

Line 36, make clear whether the number refers to the entire population of age 50+

Response

We thank the reviewer for their valuable comments on our manuscript

We corrected this to read as “Globally, this number is expected to increase to an estimated 7.5 million by 2020. Eastern and Southern Africa being the epicenter of this epidemic with an estimated 2.9 million PLHIV 50 years and older; line 35-37.

Line 53-54, could the authors add more reasons of non-disclosure for this age group? Particularly related with family separation, job loss, etc….

Response

We found only one study that reported the fear of separation as a reason fon non-disclosure- Older adults in a study conducted in Uganda delayed disclosure for fear that their spouse/partner would leave them if their positive status was known; line 54-57

Methods

Line 82, what was the purposive criteria? 15/20 were females which disproportionately seem to have the lived experiences by gender, for example.

Response

In an attempt to attain maximum variability as a criterion for purposive sampling data were collected over a period of four months and still we could not find male participants who met the inclusion criteria. To be eligible for participation in the study the participants had to be diagnosed with HIV within five years at the time of the study, aged 50 years and above, and not critically ill.  The health seeking behaviour of males disproportionally affect the gender distribution of patients seeking health, particularly for HIV services; line.

Line 94-98, How to manage investigator bias as the lead author did the interview?

Response

The lead author attended training in qualitative research as part of her fulfilment of her MPH degree. She was trained on bracketing, which is a setting aside of what we already know about the phenomena under inquiry. Therefore, she maintained a reflective attitude throughout the research process reducing the chances of her personal believes about the phenomenon to bias the study findings. We have added a statement on reflexivity under strategies to ensure credibility.

Line 98-102, what framework did the authors use to guide their in-depth interview? If not any limitation?

Response

We used semi-structured interviews and asked four broad key questions as outlined in Bevan (2014) In phenomenological inquiry, the questions are broad and open ended so that the subject has sufficient opportunity to express his or her view point extensively Giorgi (1997) The aim is to describe the participants’ experience in the way he or she experiences it, and not from some theoretical standpoint.

Line 125, could the authors provide some strategies to ensure credibility, transferability, dependability, and reliability of data

Response

We added additional strategies used to ensure trustworthiness, the section is more detailed and identifies the methods that we used; line 129-139

Results

Line 150, figure 2 resembles the socio-ecologic model such as access at macro (policy)-level, disclosure/stigma at meso (community)-level, etc. Was this co-incidence or the authors initially considered SEM as their framework? NB. I have raised comments related with framework earlier.

Response

This is a coincidence we did not plan to use the ecological model, the figure is just a depiction of the themes that emerge from the thematic analysis.

Line 150, figure 2, how the figure was constructed (how the themes and subthemes are interlinked) should be described somewhere.

Response

As mentioned, this is just an illustration of the themes, we could have used a table to present the themes.

Interpretation of quotes in the results section: I could not see much meta-phore interpretation of most quotes which could illustrate the deep phenomenology of the participants. I suggest the authors to interpret and add more. The journal does not have word limitation and they should use the opportunity.

Response

We used descriptive phenomenology as a design for the study, the aim of descriptive phenomenology is to describe a lived experience without attempting to give meaning to it. We apologize that at this point we cannot rethink the analysis of the data using interpretative phenomenology, which focus on interpreting and describing human experiences with the aim to reveal and interpret implicit meaning in a lived experience.

Discussion

Line 385-391: What does the low self-perception and risky sexual behaviours such as not using condom, having multiple sexual partners and low concern to HIV infection tell out to the general population? What should the different stakeholders should do?

Response

We added text to highlight the role played by in upholding the low risk perceptions of older adults and the implications for public health; line 406-410.

Line 404, any suggestion for the social isolation?

Response

We added text on how to reduce loneliness among older adults living with HIV; line 423-426

Any further limitations of investigator bias? Also, any limitation as a result of male-to-female (or vice versa) interview? Sometimes women may not describe their livid experience to men or vice versa?

Response

We addressed the issue of investigator bias explaining the use of bracketing and reflexivity. The issue and gender and qualitative research is addressed; line 500-501.

Reviewer 4 Report

If possible, please replace 'some participants' as seen in results and discussion section with actual number of the participants. e.g. line 458

Author Response

If possible, please replace 'some participants' as seen in results and discussion section with actual number of the participants. e.g. line 458

Response

Thanks for the comment. We are not able to use actual numbers because we used thematic analysis and did not quantify any of the responses of the participants except where we collected demographic data to describe the study sample.
